# Serum Cystatin C as a Biomarker for Early Diabetic Kidney Disease and Dyslipidemia in Young Type 1 Diabetes Patients

**DOI:** 10.3390/medicina58020218

**Published:** 2022-02-01

**Authors:** Ingrida Stankute, Lina Radzeviciene, Ausra Monstaviciene, Rimante Dobrovolskiene, Evalda Danyte, Rasa Verkauskiene

**Affiliations:** 1Institute of Endocrinology, Lithuanian University of Health Sciences, Eiveniu 2, LT-50009 Kaunas, Lithuania; lina.radzeviciene@lsmuni.lt (L.R.); evalda.danyte@lsmuni.lt (E.D.); rasa.verkauskiene@lsmuni.lt (R.V.); 2Medical Academy, Lithuanian University of Health Sciences, Eiveniu 2, LT-50161 Kaunas, Lithuania; ausra.morkunaite@stud.lsmu.lt (A.M.); rimante.dobrovolskiene@lsmuni.lt (R.D.)

**Keywords:** cystatin C, type 1 diabetes, microvascular complication, dyslipidemia, youth

## Abstract

*Background and objectives:* This study aimed to assess the clinical significance of serum cystatin C in the early diagnosis of renal injury and its association with dyslipidemia in young T1D patients. *Materials and Methods:* A total of 779 subjects were evaluated for kidney function by estimating glomerular filtration rate (eGFR) based on serum creatinine (eGFRcreat) and cystatin C (eGFRcys). *Results:* The median age of study subjects was 16.2 years (2.1;26.4), diabetes duration—5.3 years (0.51;24.0). The median of HbA1c was 8% (5.2;19.9) (64 mmol/mol (33.3;194)); 24.2% of participants had HbA1c < 7% (53 mmol/mol). Elevated albumin excretion rate was found in 13.5% of subjects. The median of cystatin C was 0.8 mg/L (0.33;1.71), the median of creatinine—63 µmol/L (6;126). The median of eGFRcys was lower than eGFRcreat (92 mL/min/1.73 m^2^ vs. 101 mL/min/1.73 m^2^, *p* < 0.001). A total of 30.2% of all patients were classified as having worse kidney function when using cystatin C vs. creatinine for eGFR calculation. Linear correlations were found between cystatin C and HbA1c, r = −0.088, *p* < 0.05, as well as cystatin C and HDL, r = −0.097, *p* < 0.01. *Conclusions:* This study showed that cystatin C might be used as an additional biomarker of early kidney injury in young patients with T1D.

## 1. Introduction

Diabetic kidney disease is defined as renal dysfunction or structural abnormalities lasting for 3 or more months and highly affecting the patient’s quality of life [1]. Starting with glomerular hypertrophy, hyperfiltration, and hyperperfusion, diabetic kidney damage progresses to end-stage renal disease, which is one of the leading causes of morbidity and mortality in young type 1 diabetic patients [2]. Estimated glomerular filtration rate (eGFR) below 60 mL/min/1.73 m^2^ determines chronic kidney disease. Recent data suggest the benefits of GFR calculated with serum cystatin C for evaluation of early risk for end-stage renal disease [3].

Cystatin C is a low-molecular-weight protein (Mw 13,343 Da), an inhibitor of endogenous cysteine protease [4]. It is produced in all tissues and found in all biological fluids. Through regulating cysteine protease activity, cystatin C was reported to be involved in many disease processes, such as inflammation and tumor metastasis [5]. Recent studies showed that cystatin C is an ideal endogenous glomerular filtration rate marker [6,7] and is a more sensitive parameter for renal function assessment compared to serum creatinine [4].

Increased cystatin C and disorders of lipid metabolism are associated with a higher risk of cardiovascular diseases (CVD), which are the leading cause of mortality in patients with type 1 diabetes (T1D) [7,8,9]. Early observations of lipid profile in patients with T1D, revealing pro-atherogenic features, such as hypercholesterolemia and hypertriglyceridemia, are associated with poor glycemic control and kidney damage [8,9,10].

Recent studies showed that cystatin C is a valuable marker for acute kidney injury in the pediatric population and even helps to stratify CVD risk in children with chronic kidney disease [11,12]. However, in the SEARCH for Diabetes in Youth study, higher levels of cystatin C were found in healthy subjects compared to type 1 and type 2 diabetes patients [13].

Since there are controversies regarding the role of cystatin C in the assessment of kidney function, we aimed to assess the clinical significance of serum cystatin C in the early diagnosis of renal injury and its association with dyslipidemia in young T1D patients.

## 2. Materials and Methods

### 2.1. Study Design and Subjects

A cross-sectional population-based study was carried out in Lithuanian University of Health Sciences, Department of Endocrinology, the whole project is described previously [14]. This particular study is a part of the whole project, here, we included 531 children and 248 adults younger than 26 years of age with T1D diagnosis in Lithuania. All patients had been confirmed with T1D diagnosis with at least one positive islet autoimmune marker: glutamic acid decarboxylase 65-kilodalton isoform, tyrosine phosphatase-related islet antigen 2 and/or insulin autoantibodies; and were treated with insulin. We included patients with disease duration more than 6 months because of known metabolic instability at the onset of diabetes [15].

Detailed clinical assessment and examination of this cohort were described previously [16], briefly at the study entry, data on the onset, duration, and treatment of diabetes were collected, clinical examination included height (cm), weight (kg), waist circumference (cm) and body mass index (BMI) (kg/m^2^). BMI values were converted to Z-scores and stratified into 4 weight status groups: underweight, normal weight, overweight, and obesity, using the World Health Organization approach [17]. Waist circumference (cm) divided by height (cm) values were used to evaluate Waist-to-Height ratio (WtHR): optimal WtHR was considered <0.5 [18].

Diabetes control was assessed by glycated hemoglobin (HbA1c). Fasting blood samples were taken for analysis of lipid profile, creatinine, and cystatin C. All patients were screened for diabetic retinopathy and neuropathy, and the level of albuminuria in 24-h urine was assessed.

### 2.2. Laboratory Analyses

HbA1c, lipids, and serum creatinine were measured using UniCel DxC 800 Synchron system (Beckman Coulter, Brea, CA, USA). Normal cut-off values for HbA1c were 4–6% (20 mmol/mol–42 mmol/mol). Optimal metabolic control was defined as HbA1c below 7% (53 mmol/mol) without severe hypoglycemias as recommended by International Society for Pediatric and Adolescent Diabetes (ISPAD) [19]. Normal values for low-density lipoprotein cholesterol (LDL), high-density lipoprotein cholesterol (HDL), and triglycerides (Tg) were defined as <2.6 mmol/L, >1.1 mmol/L and <1.7 mmol/L, respectively. Normal values for total cholesterol (TCh) were <5.2 mmol/L for patients ≥16 years and <5.5 mmol/L for children under 16 years. At least one abnormal lipid level indicated dyslipidemia. Normal cut-off values for creatinine were 39–91 µmol/L. Cystatin C was measured by an automated enzyme immunoassay system AIA 2000 (TOSOH Corporation, Tokyo, Japan) with a normal range of 0.52–0.97 mg/L.

### 2.3. Evaluation of Kidney Function

The estimated glomerular filtration rate (eGFR) values were calculated using serum creatinine (eGFRcreat) and serum cystatin C (eGFRcys) for all participants. Creatinine-based Schwartz formula [20] for eGFRcreat was used to calculate eGFRcreat in children. For eGFRcreat and eGFRcys in adults and eGFRcys in children, equations from the Chronic Kidney Disease Epidemiology Collaboration were used [21,22].

Kidney function was staged according to American Diabetes Association (ADA) guidelines [23]: Level 1 if GFR ≥ 90 mL/min./1.73 m^2^; Level 2—GFR 60–89 mL/min/1.73 m^2^; Level 3—GFR 30–59 mL/min/1.73 m^2^; Level 4—GFR 15–29 mL/min/1.73 m^2^ and Level 5—GFR < 15 mL/min/1.73 m^2^.

Furthermore, we classified patients according to eGFRcreat and eGFRcys levels: Group 1—if eGFRcreat level was equal to eGFRcys level; Group 2—if eGFRcreat level < eGFRcys level; Group 3—if eGFRcreat level > eGFRcys level.

### 2.4. Evaluation of Microvascular Complications

Digital fundus photos were taken and evaluated by a single ophthalmologist specialized in diabetic retinopathy.

All patients were screened for the level of albuminuria in 24-h urine as reported earlier [24]. Normal values were determined if the albumin excretion rate (AER) was below 30 mg/24 h; moderately increased albuminuria—between 30 and 300 mg/24 h; significantly increased albuminuria—exceeding 300 mg/24 h.

All patients were screened for the presence of symptoms or signs of symmetrical peripheral neuropathy. We applied Michigan Neuropathy Screening Questionnaire and evaluated the sensations of vibration, pressure, and temperature. Diagnosis of peripheral neuropathy was confirmed if two or more of these tests were abnormal [25,26].

### 2.5. Statistical Analyses

IBM SPSS Statistics software was used for statistical analyses. Student’s 2-tailed *t*-test, χ^2^ statistics, parametric one-way ANOVA (for normally distributed data), and Mann–Whitney U-test (for nonnormal data distribution) or Kruskal–Wallis one-way ANOVA (in the case of ordinal data) were used. Pearson correlation coefficient (for normally distributed data), and Spearman’s coefficient (for nonnormal data) were used while testing associations between continuous variables. *p*-values < 0.05 were considered statistically significant. All *p*-values were 2-tailed.

## 3. Results

### 3.1. General Characteristics of the Cohort

Overall, 779 patients with T1D were included in the study, 51.2% (*n* = 399) were females. The median of age at inclusion was 16.2 years (2.1;26.4), age at the onset of T1D—9.4 years (0.8;24.9), the median of diabetes duration—5.3 years (0.51;24.0). There were 51.9% (*n* = 404) of patients with diabetes duration ≥ 5 years.

Normal BMI was found in 75.3% (*n* = 581) of participants, 20.1% (*n* = 155) were overweight, 3.6% (*n* = 28)—obese and 1% (*n* = 8)—underweight. Optimal WtHR was found in 85.3% (*n* = 622) of participants.

Optimal glycemic control was found in 24.2% (*n* = 188) of all patients. Higher than normal serum cystatin C concentration was present in 10.8% (*n* = 84) of all patients.

General characteristics and comparisons of children vs. adult groups and males vs. females are presented in Table 1.

### 3.2. Dyslipidemia and Microvascular Complications

There were 26.1% of patients with at least one microvascular complication. Retinopathy was diagnosed in 9% of patients in the whole cohort, neuropathy in 10.8%. A total of 13.5% were found to have elevated AER, 49.5% of them had moderately increased albuminuria. The frequency of microvascular complications, dyslipidemia, and obesity are presented in Table 2.

Patients with diagnosed retinopathy, neuropathy, and moderately increased albuminuria had worse glycemic control than those without these complications, HbA1c 9.6% vs. 7.9%, 8.8% vs. 7.9%, and 8.6% vs. 8%, respectively, all *p*-values < 0.05. Patients with dyslipidemia had higher HbA1c than patients with normal lipid profiles, 8.3% vs. 7.6%, respectively, *p* < 0.001.

The duration of diabetes was significantly directly related to HbA1c, creatinine, AER, TCh, LDL, and Tg concentrations. Negative linear correlations were found between cystatin C and HbA1c, r = −0.088, *p* < 0.05, as well as cystatin C and HDL, *r* = −0.097, *p* < 0.01. HbA1c correlated directly with TCh, LDL, Tg, eGFRcreat, and eGFRcys, *p* < 0.001, and negatively with HDL, *p* < 0.05. All correlations are presented in Table 3; the correlation between Cystatine and eGFRcys as well as Creatinine and eGFRcreat are presented in Figure 1.

### 3.3. GFR and Kidney Function

The median of eGFRcys was lower in the whole cohort compared to eGFRcreat, 92 (57;201) vs. 101 (51;194) mL/min/1.73 m^2^, respectively, *p* < 0.001. Statistically significant difference was found in the children group comparing eGFRcreat vs. eGFRcys, 97 (51;169) vs. 87 (57;201) mL/min/1.73 m^2^, *p* < 0.001.

The grouping of all patients by kidney function level using either eGFRcreat or eGFRcys is presented in Figure 2. There were more patients classified with level 2 kidney function when using cystatin C vs. creatinine for eGFR calculations, 44.7% vs. 23.8%, respectively, *p* < 0.001.

After grouping patients according to the equivalency of eGFR levels calculated by creatinine and cystatin C, we found that 30.2% of patients had worse eGFR levels calculated using cystatin C (Group 2 when eGFRcreat level < eGFRcys level), 61.1% had the same eGFR level using both biomarkers (Group 1) and 8.7% were classified as Group 3 (when eGFRcreat level > eGFRcys level).

## 4. Discussion

End-stage renal disease in diabetes patients is known to be the leading cause of increased risk for premature mortality [27]. Only having clear and reliable biomarkers for early diabetic kidney damage detection will lead to early intervention [28]. In the present study, we assessed the value of serum cystatin C as a biomarker for diabetic kidney injury in children and young adults with T1D diagnoses.

The main finding of our study was that using cystatin C helped to find those young T1D patients who may be suffering from early kidney damage, as one-third of the whole cohort was classified with worse eGFR levels when using cystatin C vs. creatinine. As demonstrated in the Tsai et al. study where the adult population using eGFR based on cystatin C helped to reclassify kidney function from preserved to reduced, especially in patients with diabetes [29], this means cystatin C could be used as an earlier predictor of kidney damage compared to creatinine. There are few studies of the prognostic value of serum cystatin C in children with diabetes, yet most studies evaluated cystatin C in children with acute kidney injury. The 2017 meta-analysis of Nakhjavan-Shahraki et al. showed that serum cystatin C is a potentially more sensitive marker of acute kidney damage in children compared to serum creatinine [11].

It is known that the pathogenesis of diabetic kidney damage starts with glomerular hypertrophy, hyperfiltration and hyperperfusion state [2], therefore it is important to find a reliable and resistant biomarker of early kidney injury in this phase. Huang et al.’s study showed that eGFR was based on low-molecular-weight proteins, as cystatin C was not altered by hyperfiltration on the children cohort [30]. Combining these results with our findings increases the importance of cystatin C usage in clinical practice.

Leem et al. demonstrated that cystatin C is not only a reliable biomarker but also is associated with the recovery prognosis of acute kidney injury in critically ill adults [31]. Furthermore, Murty et al. found that cystatin C is superior over creatinine for GFR estimation in early kidney injury stages, especially in the so-called creatinine blind GFR area (40–70 mL/min/1.73 m^2^) [32], though this study was performed on an adult cohort, the findings could substantiate our results where we found that using cystatin C could help to capture young diabetes patients in early kidney injury phase when GFR is between 60 and 89 mL/min/1.73 m^2^.

Consistent with the Third National Health and Nutrition Examination Survey (NHANES III), our study showed that the levels of serum cystatin C were higher in children compared to adults [33] and were significantly higher in males compared to females in all age groups [34,35]. On the other hand, Norlund et al. did not find any differences in cystatin C levels between genders [36]. These discrepancies might be explained by the different age distribution of the studied cohorts: the median age of our diabetic cohort was 16.2 years, while Norlund et al. included healthy adult patients above the age of 20 years.

A negative correlation between serum cystatin C and HbA1c was found in our investigation, which is in agreement with the SEARCH for Diabetes in Youth Study and the results from Maahs et al.’s study [13,35], however, the importance of these findings in clinical practice remains unclear. Furthermore, our study showed a linear negative correlation between cystatin C and HDL cholesterol. A low HDL level is used as one of the criteria for the clinical diagnosis of metabolic syndrome [37]. A cross-sectional study on 925 dyslipidemic patients showed a progressive increase in cystatin C with the increasing number of metabolic syndrome components [38]. Patients with higher levels of cystatin C were shown to be at two-fold increased risk of cardiovascular events even after adjusting for a large variety of potential confounders [39]. A recent meta-analysis of over 22,000 participants from 14 studies showed that higher cystatin C levels were strongly and independently associated with specific endpoints such as stroke, myocardial infarction, and heart failure [40].

One of the limitations of our study is that we did not measure GFR as it is considered to be the best assay of kidney function [41]. However, it would be clinically and ethically difficult to perform, especially in children, as it requires intravenous injection of the filtration marker. Therefore, most of the contemporary studies use estimated GFR levels for the evaluation of kidney function, either in children or in adults [41].

The other challenge that our study faced was the lack of pediatric normative values for serum cystatin C. However, this is a worldwide issue, even for the most frequently used biomarkers, such as neutrophil gelatinase-associated lipocalin, kidney injury molecule-I, and serum cystatin C, there are insufficient reference values in the pediatric population [28].

## 5. Conclusions

Serum cystatin C adds significance for diagnosing early kidney damage, as our study showed it could be successfully used in young patients with T1D for estimating GFR. To assess the precise prognostic value of cystatin C, we need further follow-up studies of these young T1D patients to identify those who will develop advanced diabetic kidney disease.

## Figures and Tables

**Figure 1 medicina-58-00218-f001:**
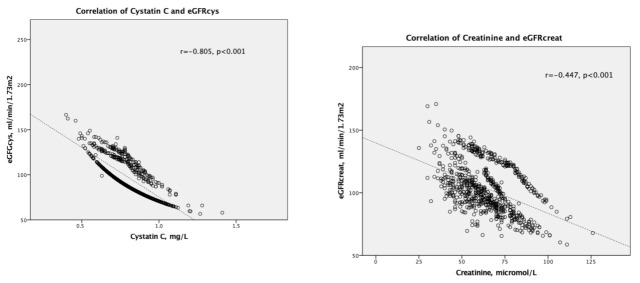
Correlation of cystatin C and estimated GFR based on cystatin C (eGFRcys) (**The Left Picture**); correlation of creatinine and estimated GFR based on creatinine (eGFRcreat) (**The Right Picture**).

**Figure 2 medicina-58-00218-f002:**
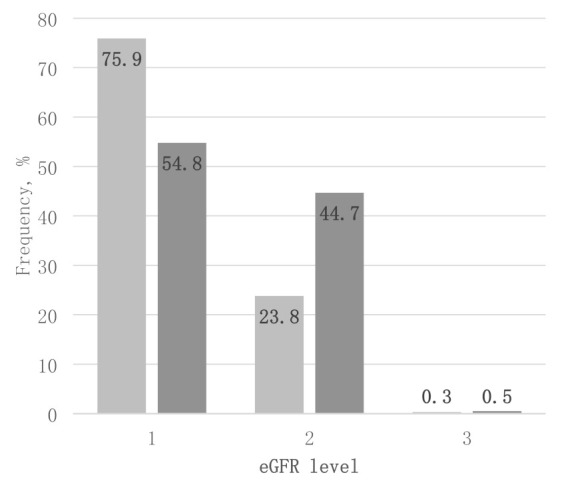
The frequency of estimated glomerular filtration rate (eGFR) levels based on serum creatine (eGFRcreat) and serum cystatin C (eGFRcys) in the whole cohort. Level 1 when GFR ≥ 90 mL/min/1.73 m^2^; 2—GFR 60–89 mL/min/1.73 m^2^; 3—GFR 30–59 mL/min/1.73 m^2^.

**Table 1 medicina-58-00218-t001:** General characteristics of study subjects.

Category	Gender/Total	Median (Range), Total Cohort	Age Groups		*p*-Value (Comparing Children vs. Adults
			Median (Range) in Children (<18 Years)	Median (Range) in Adults (≥18 Years)	
BMI Z-score	Males	0.23 (−3.66;4.34)	0.32 (−3.66;4.34)	0.07 (−1.91;3.16)	NS
	Females	0.36 (−3.96;2.97)	0.36 (−3.96;2.8)	0.37 (−1.87;2.97)	NS
	Total	0.29 (−3.96;4.34)	0.34 (−3.96;4.34)	0.29 (−1.91;3.16)	NS
WtHR	Males	0.44 (0.36;0.8)	0.44 (0.38;0.8)	0.44 (0.36;0.61)	NS
	Females	0.45 (0.25;0.63)	0.44 (0.25;0.59)	0.45 0.37;0.63)	NS
	Total	0.44 (0.25;0.8)	0.44 (0.25;0.8)	0.45 (0.36;0.63)	NS
HbA1c, % mmol/mol	Males	7.8 (5.2;14.8) ^a^ 61.7 (33.3;138.3)	7.8 (5.3;14.8) ^b^ 61.7 (34.4;138.3)	7.9 (5.2;14.1) 62.8 (33.3;138.3)	0.047
	Females	8.3 (5.4;19.9) ^a^ 67.2 (35.5;194)	8.3 (5.5;19.9) ^b^ 67.2 (35.5;194)	8.3 (5.4;14.1) 67.2 (25.5;194)	NS
	Total	8 (5.2;19.9) 63.9 (33.3;194)	8 (5.3;19.9) 63.9 (34.4;194)	8.2 (5.2;14.1) 66.1 (33.3;130.6)	NS
Serum creatinine, μmol/L *	Males	70 (7;126) ^a^	62 (7;107) ^b^	80 (35;113) ^c^	<0.001
	Females	59 (6;111) ^a^	58 (6;94) ^b^	64 (36;111) ^c^	<0.001
	Total	63 (6;126)	59 (6;107)	70 (35;113)	<0.001
Serum cystatine C, mg/L	Males	0.84 (0.08;1.71) ^a^	0.85 (0.54;1.27) ^b^	0.82 (0.08;1.71) ^c^	NS
	Females	0.78 (0.33;1.19) ^a^	0.80 (0.33;1.11) ^b^	0.76 (0.52;1.19) ^c^	0.004
	Total	0.81 (0.08;1.71)	0.83 (0.33;1.27)	0.79 (0.08;1.71)	0.008
Total cholesterol, mmol/L	Males	4.6 (1.9;8.9) ^a^	4.7 (1.9;8.9) ^b^	4.6 (2.9;7.9) ^c^	NS
	Females	4.9 (2.5;9.3) ^a^	4.9 (3.1;8.5) ^b^	5.0 (2.5;9.3) ^c^	NS
	Total	4.8 (1.9;9.3)	4.8 (1.9;8.9)	4.7 (2.5;9.3)	NS
LDL-cholesterol, mmol/L	Males	2.6 (0.5;5.8) ^a^	2.5 (0.5;5.8) ^b^	2.6 (1.3;5.7) ^c^	NS
	Females	2.8 (0.8;6.5) ^a^	2.8 (1.1;6.1) ^b^	2.8 (0.8;6.5) ^c^	NS
	Total	2.7 (0.5;6.5)	2.7 (0.5;6.1)	2.7 (0.8;6.5)	NS
HDL-cholesterol, mmol/L	Males	1.4 (0.6;2.7) ^a^	1.4 (0.8;2.7)	1.3 (0.6;2.3) ^c^	<0.001
	Females	1.5 (0.3;2.6) ^a^	1.5 (0.6;2.6)	1.5 (0.3;2.6) ^c^	NS
	Total	1.4 (0.3;2.7)	1.5 (0.6;2.7)	1.4 (0.3;2.6)	0.012
Triglycerides, mmol/L	Males	0.7 (0.1;5.3) ^a^	0.7 (0.1;5.3) ^b^	0.9 (0.2;5.3)	<0.001
	Females	0.8 (0.3;5.1) ^a^	0.8 (0.3;5.1) ^b^	0.9 (0.3;3.3)	NS
	Total	0.8 (0.1;5.3)	0.7 (0.1;5.3)	0.9 (0.2;5.3)	<0.001
AER, mg/24 h	Males	7.5 (0.06;787)	7 (1;787)	9 (0.06;533)	NS
	Females	6.8 (0.08;667)	5.6 (0.5;180)	9 (0.08;667)	<0.001
	Total	7.2 (0.06;787)	6 (0.5;787)	9 (0.06;667)	<0.001
eGFRcreat, ml/min/1.73 m^2^	Males	96 (51;194) ^a^	92 (51;169) ^b^	121 (68;194)	<0.001
	Females	104 (59;154) ^a^	98 (64;151) ^b^	115 (59;154)	<0.001
	Total	101 (51;194)	96 (51;169)	118 (58;194)	<0.001
eGFRcys, ml/min/1.73 m^2^	Males	89 (57;160) ^a^	83 (57;125) ^b^	118 (58;160)	<0.001
	Females	95 (64;201) ^a^	88 (64;201) ^b^	117 (66;146)	<0.001
	Total	92 (57;201)	85 (57;201)	117 (58;160)	<0.001

BMI, body mass index; WtHR, weight to height ratio; HbA1c, glycosylated hemoglobin; LDL, low-density lipoprotein; HDL, high-density lipoprotein; AER, albumin excretion rate, eGFRcreat, estimated glomerular filtration rate using serum creatinine; eGFRcys, estimated glomerular filtration rate using serum cystatin C; NS, not significant. * adjusted for gender, age, and BMI Z-score; a—*p* ≤ 0.001 comparing males vs. females in the whole cohort; b—*p* < 0.001 comparing males vs. females in the children group; c—*p* < 0.001 comparing males vs. females in adults’ group.

**Table 2 medicina-58-00218-t002:** Frequency of microvascular complications, dyslipidemia and obesity.

	Gender/Total	Frequency	Diabetes Duration		*p*-Value (Comparing Diabetes Duration Groups)
			<5 Years	≥5 Years	
Retinopathy, % (*n*)	Males	5 (19) ^a^	0	10.4 (19)	<0.001
	Females	12.8 (51) ^a^	1.7 (3)	21.7 (48)	<0.001
	Total	9 (70)	0.8 (3)	16.6 (37)	<0.001
Neuropathy, % (*n*)	Males	8.9 (33)	2.6 (5)	16.1 (28)	<0.001
	Females	12.6 (49)	2.3 (4)	21.2 (45)	<0.001
	Total	10.8 (82)	2.4 (9)	18.9 (73)	<0.001
Elevated AER, % (*n*)	Males	13.2 (50)	13.7 (27)	12.6 (23)	NS
	Females	13.8 (55)	10.1 (18)	16.7 (37)	NS
	Total	13.5 (105)	12 (45)	14.9 (60)	NS
Dyslipidemia, % (*n*)	Males	60.3 (229)	56.3 (111)	64.5 (118)	NS
	Females	66.2 (264)	61.8 (110)	69.7 (154)	NS
	Total	63.3 (493)	58.9 (221)	67.3 (272)	0.015
Obesity, % (*n*)	Males	3.7 (14)	4.7 (9)	2.7 (5)	NS
	Females	3.5 (14)	3.4 (6)	3.6 (8)	NS
	Total	3.6 (28)	4.1 (15)	3.2 (13)	NS

AER, albumin excretion rate; NS, not significant. ^a^—*p* < 0.001 comparing males vs. females in the whole cohort.

**Table 3 medicina-58-00218-t003:** Correlation between continuous variables.

	Duration of Diabetes, Years	BMI Z-Score	WtHR	HbA1c, %	Cystatin C, mg/L	Creatinine, μmol/L	AER, mg/24 h	Total Cholesterol, mmol/L	LDL, mmol/L	HDL, mmol/L	Triglycerides, mmol/L	eGFRcreat, ml/min/1.73 m^2^	eGFRcys ml/min/1.73 m^2^
Duration ofdiabetes, years	1.00	0.052	−0.006	0.301 ***	−0.035	0.288 ***	0.139 ***	0.089 *	0.085 *	−0.057	0.241 ***	0.148 ***	0.239 ***
BMI Z-score		1.00	0.661 ***	0.033	0.006	0.056	0.035	0.07	0.09 *	−0.044	0.089 *	−0.062	−0.019
WtHR			1.00	0.084 *	−0.096 *	−0.131 ***	−0.043	0.119 **	0.108 **	−0.028	0.154 ***	0.035	0.084 *
HbA1c, %				1.00	−0.088 *	−0.049	0.067	0.227 ***	0.211 ***	−0.083 *	0.418 ***	0.145 ***	0.108 **
Cystatin C, mg/L					1.00	0.228 ***	0.011	−0.036	−0.001	−0.097 **	0.05	−0.234 ***	−0.805 ***
Creatinine, μmol/L						1.00	0.131 ***	−0.111 **	−0.053	−0.217 ***	0.149 ***	−0.447 ***	0.023
AER, mg/24 h							1.00	0.024	0.04	−0.036	0.108 **	0.072	0.063
Total cholesterol, mmol/L								1.00	0.857 ***	0.357 ***	0.351 ***	0.062	0.017
LDL, mmol/L									1.00	0.003	0.368 ***	0.033	−0.010
HDL, mmol/L										1.00	−0.228 ***	0.000	−0.002
Triglycerides, mmol/L											1.00	0.092 *	0.052
eGFRcreat, ml/min/1.73 m^2^												1.00	0.455 ***
eGFRcys, ml/min/1.73 m^2^													1.00

BMI, body mass index; WtHR, weight to height ratio; HbA1c, glycated hemoglobin; AER, albumin excretion rate; LDL, low-density lipoprotein; HDL, high-density lipoprotein; eGFRcreat, estimated glomerular filtration rate using serum creatinine; eGFRcys, estimated glomerular filtration rate using serum cystatin C. * *p*-value < 0.05; ** *p*-value < 0.01; *** *p*-value < 0.001.

## Data Availability

All data generated or analyzed during this study are included in this published article.

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
