# Peer review of "Serum Cystatin C as a Biomarker for Early Diabetic Kidney Disease and Dyslipidemia in Young Type 1 Diabetes Patients"

_medicina, 2022, doi:10.3390/medicina58020218_

Round 1

Reviewer 1 Report

1. It is very much necessary to find a very sensitive and sophisticated diagnostic marker to diagnose renal dysfunction in pediatric nephropathy patients. Few studies have already been reported earlier about the significance of adapting serum systatin C as a marker in non-Lithuanian population. It is mandatory to refer to those studies and discuss how and what way the present is similar, differ or better than earlier evaluations.

Please refer to those studies (for eg:tps://pubmed.ncbi.nlm.nih.gov/23814415/ & https://pubmed.ncbi.nlm.nih.gov/28332367/

Please clarify, is the present study is a part of ref 14 or a separate study. It is not clear.

2. "Optimal glycemic control was found in 24.2% (n=188) of patients. Higher than normal serum cystatin C concentration was present in 10.8% (n=84) of patients". It is not clear from your results (1st paragraph) whether these 84 patients are subset of above 188, or from total patients.  Please clarify. According to your data 75% of patients had abnormal glycemic levels whereas 89% might had normal cystatin C concentration. 

3. In the figure 1, I have difficulty in interpreting the data based on description of results and results.  The values shown in figure and and description are not corroborating. Furthermore, description of eGFR levels also need to be descriptive.

4. Correlation between Cystatin C values and kidney function parameters (AER/eGFR) can be presented as a figure (rather than table) and it is more better way of presentation.

Author Response

Dear Reviewer 1, 

Thank you for your valuable insights and significant comments. Please find our responses here and see the attachment. 

  1. It is very much necessary to find a very sensitive and sophisticated diagnostic marker to diagnose renal dysfunction in pediatric nephropathy patients. Few studies have already been reported earlier about the significance of adapting serum systatin C as a marker in non-Lithuanian population. It is mandatory to refer to those studies and discuss how and what way the present is similar, differ or better than earlier evaluations.

Please refer to those studies (for eg:tps://pubmed.ncbi.nlm.nih.gov/23814415/ & https://pubmed.ncbi.nlm.nih.gov/28332367/

Response: We have definitely discussed on the findings of suggested studies in the discussion section. Also, added both as references. We believe that these insights augmented our findings.

Please clarify, is the present study is a part of ref 14 or a separate study. It is not clear.

Response: We have clarified that this study was a part of population-based research project “Genetic Diabetes in Lithuania”, it can be found in the “Materials and Methods” section. This study has the same cohort as the whole project, although with specific aim to test cystatin C as a biomarker for kidney injury in young diabetes patients.

  1. "Optimal glycemic control was found in 24.2% (n=188) of patients. Higher than normal serum cystatin C concentration was present in 10.8% (n=84) of patients". It is not clear from your results (1st paragraph) whether these 84 patients are subset of above 188, or from total patients.  Please clarify. According to your data 75% of patients had abnormal glycemic levels whereas 89% might had normal cystatin C concentration. 

Response: We have clarified that these percentages are based on all patients, it can be found in the “Results” section.

  1. In the figure 1, I have difficulty in interpreting the data based on description of results and results.  The values shown in figure and and description are not corroborating. Furthermore, description of eGFR levels also need to be descriptive.

Response: In order to make these results clearer we have described them in a separate paragraph and added clarifying information in the Figure.

  1. Correlation between Cystatin C values and kidney function parameters (AER/eGFR) can be presented as a figure (rather than table) and it is more better way of presentation.

Response: We have added two graphs, first one for correlation of cystatine C and eGFR, the second one – for creatinine and eGFR. We believe that this visualization will make it clearer for the reader.

Reviewer 2 Report

This is a nice study regarding the significance of the serum cystatin C in young cohort of patients. I suggest some minor corrections and enhancements, in order to make the manuscript more appealing and appropriate.

  1. I strongly suggest using the internationally accepted nomenclature. I suggest using the term "diabetic kidney disease", as diabetic nephropathy is just a part of the disease spectrum. Also, appropriate explanation is necessary, as in young type1 patients, deteriorated kidney function is probably the consequence of diabetic nephropathy per se. Also, terms "micro-! and "macroalbuminuria" are obsolete, please use terms as moderately and significantly increased albuminuria (ckd_compendium_fin_2_web.pdf (diabetes.org). 
  2. It is necessary to make a statement, whether the study population was receiving any additional therapy of any kind, particularly ACE inhibitors/sartans and lipid lowering therapy.
  3. In the "introduction", please mention the hyperfiltration effect of the pathogenesis in DKD and its effect on serum cystatin C nad in the "discussion", please address this issue in regard to the results. 

Best regards!

Author Response

Dear Reviewer 2, 

Than you for your thoughtful insights and important comments. Please find our responses below and see the attachment. 

This is a nice study regarding the significance of the serum cystatin C in young cohort of patients. I suggest some minor corrections and enhancements, in order to make the manuscript more appealing and appropriate.

1. I strongly suggest using the internationally accepted nomenclature. I suggest using the term "diabetic kidney disease", as diabetic nephropathy is just a part of the disease spectrum. Also, appropriate explanation is necessary, as in young type1 patients, deteriorated kidney function is probably the consequence of diabetic nephropathy per se. Also, terms "micro-! and "macroalbuminuria" are obsolete, please use terms as moderately and significantly increased albuminuria (ckd_compendium_fin_2_web.pdf (diabetes.org). 

Response 1: We totally agree with the reviewer‘s suggestions and have corrected our manuscript according to the internationally accepted nomenclature.

2. It is necessary to make a statement, whether the study population was receiving any additional therapy of any kind, particularly ACE inhibitors/sartans and lipid lowering therapy.

Response 2: Our patients were not receiving any of the mentioned drugs. In clinical practice we  follow the ISPAD (International Society for Pediatric and Adolescent Diabetes) guidelines, therefore for elevated blood pressure and/or dyslipidemia early lifestyle interventions for at least š months were suggested as required. For patients with elevated albumin excretion rate we applied intensive follow-up, because, there are some concerns regarding the use of ACEI in young type 1 diabetes patients.

3. In the "introduction", please mention the hyperfiltration effect of the pathogenesis in DKD and its effect on serum cystatin C nad in the "discussion", please address this issue in regard to the results. 

Response 3: We have added data on hyperfiltration in the pathogenesis of DKD in the introduction section, moreover, we found new reference, which we discussed in discussion section, it even supported our main results, therefore these reviewers’ insights and comments helped to strengthen our findings.
